One pair of hands is not like another: caudate BOLD response in dogs depends on signal source and canine temperament

Cook Peter F. 1
Spivak Mark 2
Berns Gregory S. 1 gberns@emory.edu
1 Economics Department & Center for Neuropolicy, Emory University , Atlanta, GA , USA
2 Comprehensive Pet Therapy , Sandy Springs, GA , USA
Vallortigara Giorgio
Electronic publication date: 2014 Sep 30
Publication date: 2014
Volume: 2
Electronic Location ID: e596
Received 2014 Jul 3; Accepted 2014 Sep 4
Copyright: © 2014 Cook et al.
Copyright year: 2014
Copyright holder: Cook et al.
License: This is an open access article distributed under the terms of the Creative Commons Attribution License, which permits unrestricted use, distribution, reproduction and adaptation in any medium and for any purpose provided that it is properly attributed. For attribution, the original author(s), title, publication source (PeerJ) and either DOI or URL of the article must be cited.
License URL: https://creativecommons.org/licenses/by/4.0/

Keywords: fMRI, Canine cognition, Animal temperament, Caudate, Neuroimaging, Comparative neuroscience, Reward systems

Funding: Office of Naval Research N00014-13-1-0253 This work was funded by a grant from the Office of Naval Research (N00014-13-1-0253). The funders had no role in study design, data collection and analysis, decision to publish, or preparation of the manuscript.

==============================
Having previously used functional MRI to map the response to a reward signal in the ventral caudate in awake unrestrained dogs, here we examined the importance of signal source to canine caudate activation. Hand signals representing either incipient reward or no reward were presented by a familiar human (each dog’s respective handler), an unfamiliar human, and via illustrated images of hands on a computer screen to 13 dogs undergoing voluntary fMRI. All dogs had received extensive training with the reward and no-reward signals from their handlers and with the computer images and had minimal exposure to the signals from strangers. All dogs showed differentially higher BOLD response in the ventral caudate to the reward versus no reward signals, and there was a robust effect at the group level. Further, differential response to the signal source had a highly significant interaction with a dog’s general aggressivity as measured by the C-BARQ canine personality assessment. Dogs with greater aggressivity showed a higher differential response to the reward signal versus no-reward signal presented by the unfamiliar human and computer, while dogs with lower aggressivity showed a higher differential response to the reward signal versus no-reward signal from their handler. This suggests that specific facets of canine temperament bear more strongly on the perceived reward value of relevant communication signals than does reinforcement history, as each of the dogs were reinforced similarly for each signal, regardless of the source (familiar human, unfamiliar human, or computer). A group-level psychophysiological interaction (PPI) connectivity analysis showed increased functional coupling between the caudate and a region of cortex associated with visual discrimination and learning on reward versus no-reward trials. Our findings emphasize the sensitivity of the domestic dog to human social interaction, and may have other implications and applications pertinent to the training and assessment of working and pet dogs.

Introduction

The domestic dog is a rising star in behavioral neuroscience due both to his high trainability and likely co-evolution with humans over the last 10,000–30,000 years. Recently, dogs have even proved amenable to participation in awake, unrestrained neuroimaging, allowing researchers to further probe the workings of the canine mind in a non-invasive and ethical method (Berns, Brooks & Spivak, 2012). The picture that is emerging suggests dogs are highly sensitive to social context and cues, both from other dogs and from humans. Unsurprisingly for a highly social species, dogs use an extensive and nuanced vocabulary of cues in both affiliative and antagonistic interactions with conspecifics (Bradshaw & Nott, 1995; Quaranta, Siniscalchi & Vallortigara, 2007; Horowitz, 2009; Siniscalchi et al., 2013); they are perhaps more unique in their apparent interspecies sociality (Siniscalchi et al., 2010), showing sensitivity to human cues as well. It has even been suggested that dogs might serve as better comparative models for human cognition than non-human apes (Topal et al., 2009). They can perform fast mapping of novel words (Kaminski, Call & Fischer, 2004), and appear to have brain regions specialized for processing the human voice (Andics et al., 2014). Behavioral work shows that dogs can read naturalistic signals from humans that may be difficult for other species (Hare & Tomasello, 2005; Teglas et al., 2012; Gácsi et al., 2013) (although see Udell, Dorey & Wynne, 2010). Prior fMRI research in our lab has shown that the ventral caudate nucleus—a brain region known to be specialized for reward prediction and processing of positively valenced stimuli across species (Schultz, Tremblay & Hollerman, 2000; Montague & Berns, 2002; Humphries & Prescott, 2010; Daw et al., 2011)—is differentially active in dogs in response to the scent of familiar humans versus familiar dogs and unfamiliar humans (Berns, Brooks & Spivak, in press). In line with behavioral evidence (Miklosi & Topal, 2013), this suggests that, in a real sense, dogs may prefer the company of familiar humans over the company of either familiar or unfamiliar dogs.

While the amassing data are suggestive of specialized social sensitivity in the dog, the question remains to what extent simple conditioning and reinforcement history, as opposed to social specialization, can explain previous findings. To address one aspect of this question, we iterated on and expanded our earlier fMRI work with dogs. Having previously shown a replicable but heterogeneous ventral caudate response in dogs to signals from their handlers predicting either incipient food reward or no reward (Berns, Brooks & Spivak, 2012; Berns, Brooks & Spivak, 2013), here we used fMRI to examine neural responses to these same signals presented by a familiar human and an unfamiliar human and to analogous but illustrated hand signals projected on a screen (referred to forthwith as “computer” signals). Although the subject dogs had an extensive reinforcement history with the human and computer signals, we hypothesized that differential caudate BOLD response in the reward versus no-reward condition—taken here to be indicative of strength of reward prediction—would be highest with the familiar human source, and lowest with the computer, in line with the possibility that social bond, and not just food-specific reinforcement history, affects the valence of familiar cues. There is growing evidence to suggest that human–dog social bonds are durable and complex, showing much of the same nuance as those of human–human social bonds (e.g., Prato-Previde et al., 2003; Siniscalchi, Stipo & Quaranta, 2013). It is possible then that the source of a signal might indeed have a profound effect on how it is received and processed by a dog.

To examine possible effects of temperament on conditional neural response, we also collected CBARQ questionnaires (a validated method for quantifying dog temperament Duffy & Serpell, 2012) from each dog’s primary handler.

Finally, we conducted an exploratory psychophysiological interaction (PPI) analysis on the BOLD data. PPI is a connectivity measure that allows one to examine what brain areas increase functional connectivity (i.e., synchronous fluctuations of brain activity Biswal et al., 1995) with a seed region on a task or condition-specific basis. While typical BOLD contrasts used in fMRI tend to highlight regions maximally specific to the task or condition of interest, PPI can highlight more distributed network activity involving areas less specialized to the primary task (Friston et al., 1997; Rogers et al., 2007). PPI generally requires large numbers of events, so we collapsed across all source conditions (familiar human, unfamiliar human, and computer) and looked for areas with increased functional connectivity to the ventral caudate in the reward vs. no-reward conditions. This opens the possibility of uncovering task-specific reward networks across the canine brain.

Materials and Methods

Participants

Participants were dogs (n = 13) from the Atlanta community (Table 1). All were pets and/or released service dogs whose owners volunteer their time for fMRI training and experiments. All had previously completed an fMRI session in which two hand signals were presented by their primary trainer, one indicating forthcoming food reward, the other indicating no reward. Accordingly, all dogs had demonstrated an ability to remain still during training and scanning for periods of 30 s or greater. However, one dog exhibited excessive motion during this experiment and was subsequently excluded from analysis because of insufficient observations after motion censoring (see below).

Table 1 Participants.

Name	Breed	Sex	Age	Service training	
Callie	Feist	Female-spayed	4	N	
Caylin	Border Collie	Female-spayed	4	N	
Jack	Golden Retriever	Male-neutered	9	N	
Kady	Yellow Lab	Female-spayed	3	Y	
Libby	Vizsla Pit Mix	Female-spayed	7	N	
Nelson	Cairn Terrier Mix	Male-neutered	3	N	
Ohana	Golden Retriever	Female-spayed	4	Y	
Pearl	Golden Retriever	Female-spayed	3	Y	
Stella	Bouvier	Female-spayed	5	N	
Tigger	Boston Terrier	Male-neutered	6	N	
Velcro	Vizsla	Male-intact	5	N	
Zen	Yellow Lab	Male-neutered	3	Y	

This study was performed in strict accordance with the recommendations in the Guide for the Care and Use of Laboratory Animals of the National Institutes of Health. The study was approved by the Emory University IACUC (Protocol #DAR-2001274-120814BA), and all dogs’ owners gave written consent for participation in the study.

Training

For participation in previous experiments (details in: Berns, Brooks & Spivak, 2013; Berns, Brooks & Spivak, in press), dogs took part in a training program using behavior shaping, desensitization, habituation, and behavior chaining to prepare them to be comfortable with the physical confines of the MRI bore and the loud noise produced by scanning. The end goal of initial training was that each dog remain comfortable and motionless for up to 30 s at a time during live scanning.

In the current experiment, dogs received reward and no reward signals from a familiar human (their primary experimental trainer), an unfamiliar human, and from computer-generated stimuli on a projection screen. As mentioned above, each dog in this experiment was highly trained on two hand signals, an upright raised left hand signifying reward, and both hands held sideways and inward-pointing, with the back of each hand toward the dog, signifying no reward (as described in Berns, Brooks & Spivak, 2012). No further training for these signals was required.

The current experiment involved two new experimental stimuli: computer-generated images to be presented on a projection screen. These were to serve the same signaling function as the previously learned hand signals (reward and no-reward). The computer reward signal consisted of an illustrated image of a raised hand, oriented so it would appear as a raised left hand facing the dog. The computer no-reward signal consisted of two illustrated inward pointing hands with thumbs up, oriented to appear as if the backs of the hands were facing the dog (Fig. 1).

Figure 1 The computer reward (A) and no-reward (B) signals.

For maximum discriminability, the two signals were presented on black backgrounds, and each at one of the wavelengths to which dogs are most sensitive (Neitz, Geist & Jacobs, 1989), a yellow–green for the reward signal (555 nm), and a medium blue (429 nm) for the no-reward signal. Wavelengths were converted RGB values (http://rohanhill.com/tools/WaveToRGB/). Because we were not making predictions about visual processing, but rather wanted two stimuli that were maximally discriminable, we did not attempt to normalize the contrast and luminance of the two computer stimuli.

Training for the current experiment involved biweekly instruction at our training facility overseen by core project staff and regular training at home overseen by each dogs’ primary experimental trainer. Dogs were trained to associate the green raised hand signal with reward and the sideways blue hand signal with no reward via the same method previously employed to teach them the human hand signals (Berns, Brooks & Spivak, 2013). During training sessions, each dog cooperatively stationed him- or herself in a custom-made foam chinrest placed inside of a mock MRI coil. Presentation of signals was contingent on the dog’s being able to station calmly and with minimal motion. The signals were displayed on a laptop screen placed approximately 20 inches in front of the dog, and were presented in short (<20 trials) sessions. Presentation order was semi-randomized (each signal was presented no more than 4 times consecutively, and reward to no-reward ratio was between 1:1 and 1:3, with the reward signal becoming less frequent once each dog became comfortable and experienced with the protocol). If the dog was able to hold still for the duration of the green reward hand signal, a food reward was dispensed immediately after. Following the blue sideways hand signal, no reward was dispensed. If the dog left the station during either signal, the stimulus was turned off, and the handler waited until the dog returned, at which point they presented the same stimulus again.

Association learning with the two computer stimuli was conducted over multiple sessions and for an amount of time comparable to initial training with the human hand signals (approximately 1 month).

When dogs were accustomed to viewing the computer signals, further training was conducted at our training facility. First, dogs were acclimated to a new food-delivery system (dubbed the “treat-kabob”). This was necessary because, while previously food reward had been delivered directly by hand, in the current experiment the projection screen required for presentation of the computer stimuli made this difficult. Furthermore, we wished to evaluate only the context variability of the signals. Therefore, we needed to keep the food delivery methodology constant across the three signal source states. Two variants of the food-delivery system were used—one in which a long wooden dowel with a small treat placed on a dull skewer at the end was slid up a PVC pipe and presented to the dog, and another in which the same dowel was used to bring the treat to the dog’s mouth without the PVC guide tube (the specific delivery method was adjusted for the dog’s comfort). In both cases, dogs were first exposed to the treat kabob in their chin stations on the ground until it was determined they were not nervous and were competent at removing the treats from the skewer. Training was then transitioned into the mock MRI scanner (Fig. 2A, see Berns, Brooks & Spivak, 2013). In 1–3 sessions with each dog’s primary experimental trainer presenting the familiar reward and no-reward hand signals, food reward was delivered via the treat kabob by a project staff member sitting below and to the side of the bore. Each dog also received one or two sessions with an unfamiliar human delivering the reward and no-reward hand signals, to assess their willingness to remain stationary in the presence of an unfamiliar human. Note that the human used in this behavioral assessment was different than the unfamiliar human who delivered signals at testing. When dogs were judged comfortable with this setup, training commenced for acclimation to a screen being placed in the bore. A computer monitor was placed at the end of the bore, directly in front of the dog’s chin station, to simulate the projection screen used in live scanning. These training sessions were aimed at making the dogs comfortable with stationing in the bore with a visual barrier in front of them.

Figure 2 (A) Participant Kady stationed in the mock scanner with the treat kabob. (B) Participant Zen stationed in the MRI bore, facing the projection screen (and wearing ear protection).

We did not have a strict behavioral metric to determine when this training was complete—rather a practical measure was used. Dogs were cleared for testing when they had completed at least one month of regular practice and were able to do an extended “dress rehearsal” session in the mock scanner without showing anxiety or escape behavior. This session involved stationing with the monitor in place, recorded scanner noise playing, and the primary handler out of view. Dogs also had to be wearing their ear protection, and treats were delivered via the treat kabob. Total time in training, from first introduction to the computer stimuli to finishing “dress rehearsal” ranged from 8 weeks to 16 weeks, with the duration affected by both the aptitude of the dog and the diligence of the owner in implementing homework sessions.

MRI scanning

All scanning for the current experiment was conducted with a Siemens 3 T Trio whole-body scanner. Dogs were stationed in their custom chin-rests, which had been placed in a standard neck coil as previously described in Berns, Brooks & Spivak (2013) (Fig. 2B). All participants wore ear protection during scanning, either Mutt Muffs™ or ear plugs with wrap, depending on dog and owner preference.

Each scan session began with a 3 s, single image localizer in the sagittal plane (SPGR sequence, slice thickness = 4 mm, TR = 9.2 ms, TE = 4.15 ms, flip angle = 40°, 256 ×256 matrix, FOV = 220 mm).

A T2-weighted structural image was previously acquired during one of our earlier experiments using a turbo spin-echo sequence (25–30 2 mm slices, TR = 3,940 ms, TE = 8.9 ms, flip angle = 131°, 26 echo trains, 128 × 128 matrix, FOV = 192 mm), which lasted ∼30 s.

Functional scans used a single-shot echo-planar imaging (EPI) sequence to acquire volumes of 24 sequential 3 mm slices with a 10% gap (TE = 28 ms, TR = 1,400 ms, flip angle = 70°, 64 × 64 matrix, 3 mm in-plane voxel size, FOV = 192 mm). Slices were oriented dorsally to the dog’s brain (coronal to the magnet, as, in the sphinx position, the dogs’ heads were positioned 90° from the usual human orientation) with the phase-encoding direction right-to-left. Sequential slices were used to minimize between-plane offsets from participant movement, and the 10% slice gap minimized the crosstalk that can occur with sequential scan sequences.

Six runs of up to 300 functional volumes were acquired, each lasting approximately 6 min. For the first dog (Zen), three runs of up to 600 functional volumes were acquired, each lasting approximately 12 min. This was subsequently split into 6 runs in a counterbalanced fashion for the remainder of the dogs. As part of a separate experiment, an additional functional run was acquired during the session in which the dog was presented with different types of visual stimuli on the screen; however, these data are not analyzed or reported here.

During functional scanning, reward was delivered via the aforementioned treat-kabob, operated by a project staff member sitting below and to the side of the scanner bore (out of sight of the dog).

Experimental design

Stimuli for this experiment were as described above: the two natural hand signals representing reward and no-reward delivered by a familiar and an unfamiliar human, and the two illustrated computer signals representing reward and no-reward, projected on a screen at the head of the bore.

Each subject dog received 15 reward and 15 no-reward signals with the familiar human, the unfamiliar human, and with the computer images, for 90 trials overall across the three source conditions. Each stimulus was presented for approximately 10 s, regardless of source. These 90 trials were broken into six runs of 15 trials each. An event-based design was used, with reward and no-reward trials presented semi-randomly within each run (either 7 reward trial and 8 no-reward trials or vice versa, and with no more than three of either stimulus type presented consecutively). The six runs for each dog were always in the same order: familiar human, unfamiliar human, computer, computer, unfamiliar human, familiar human (as part of a different experiment, an additional run was included in the middle of this sequence in which different visual stimuli were presented on the computer screen and are analyzed and reported elsewhere). Low sample size didn’t allow for different counterbalancing across participants or randomizing order, and running each source condition on a separate day to avoid effects from in-session habituation was not practical. This ABCCBA pattern of presentation controlled, at least partially, for effects in the BOLD signal from habituation, sensitization, and scanner drift becoming confounded with signal source. In addition, we determined that placing the computer runs first or last might be difficult for some dogs—in practice we had observed that dogs tended to move more and show more signs of anxiety when receiving computer signals as opposed to receiving signals directly from their handlers. By placing the computer signals in the middle of the experimental sequence, each dog had a chance to “warm up” to the task with the more familiar source conditions before the computer runs, but also had the more familiar (potentially easier) source conditions at the end of the experimental sequence, when stress and fatigue may have made them less inclined to continue participating. The first dog’s 90 trials were broken instead into three runs of 30 trials each (15 reward and 15 no-reward), in the following order: familiar, unfamiliar, computer. Thus, he received the same amount of total trials and trials for each condition as the other dogs, but without the pyramidal order provided the other dogs.

Behavioral criteria during testing for all dogs were the same as during training—each dog was required to hold still for the duration of each signal. Following a reward signal, they received a small piece of hot-dog via the treat-kabob. No reward was given following a no-reward signal. Following each 15-trial run, the dog was taken out of the scanner and allowed to walk around, drink water, etc.

For live scanning, some dogs did show some anxiety when first placed in the bore with the projection screen in place and their handler out of sight. Basic training mechanisms were employed to work around this, reprising the initial acclimation approach taken in the mock scanner at our training facility discussed previously. In brief, prior to the beginning of these scanner runs, the dogs were stationed with the screen present and their owner’s face in view. They were rewarded a number of times as the owner was approximated back and to the side, and then live scanning began.

Event recording

Trial events (onset and offset of reward and no-reward signals) were recorded by an observer via a four-button MRI-compatible button-box. The observer stood next to the experimental trainer and unfamiliar person respectively on runs 1, 2, 5, and 6, such that they could see the dog’s head in the bore of the magnet. On these runs, the observer signaled the experimental trainer and unfamiliar person when to present which signal. On runs 3 and 4, the computer runs, the observer moved to the side so that the dog was unable to see them, but the they could just see the very tip of the dog’s nose (to ascertain that the dog was still appropriately stationed/not moving). On the computer runs, the observer used the button-box to present and advance the reward and no-reward stimuli (while simultaneously recording onset/offset).

A laptop computer running Matlab (MathWorks) and Cogent (FIL, University College London) was connected to the button-box via serial port, and recorded both the button-box responses by the observer and scanner sequence pulses.

C-BARQ

Because we hypothesized that variability in participant disposition toward handlers and strangers could have a strong impact on the relative reward value of the two signal types across different source conditions the dogs’ handlers completed the Canine Behavioral Assessment & Research Questionnaires (C-BARQ). These have been used on over 20,000 dogs and represent a standardized and validated (Duffy & Serpell, 2012) tool for obtaining behavioral measures from owner report. The C-BARQ consists of 101 questions asking the respondent to report on how the dog typically responds to common events.

Scores between 0 (minimum) and 5 (maximum) are then computed for 14 behavioral categories. Because of the small number of dogs relative to the number of C-BARQ factors, there was a high potential for factors to be correlated with each other. This collinearity, combined with the number of factors, would be problematic in any modeling of the neural data. Therefore, we performed principal component analysis (PCA) on the 14 C-BARQ scores from our subjects, with a limit of four factors. By limiting to four factors, we were able to include these factors in the neural model without overfitting, while still being able to identify the major groupings of C-BARQ dimensions that were responsible for differences in neural activation.

Functional data preprocessing and analysis

Preprocessing was conducted using AFNI (NIH) and its associated functions, and most steps were identical to those listed in Berns, Brooks & Spivak (in press). In brief, 2-pass, 6-parameter affine motion correction was used with a hand-selected reference volume for each dog. Because dogs moved between trials (and when rewarded), aggressive censoring was carried out, relying on a combination of outlier voxels in terms of signal intensity and estimated motion. Censored files were inspected visually to be certain that bad volumes (e.g., when the dog’s head was out of the scanner) were not included. The majority of censored volumes followed the consumption of food. On average, 51% of total EPI volumes were retained for each subject (ranging from 38% to 60%). This was in line with previous experiments using reward and no-reward signals (Berns, Brooks & Spivak, 2013; Berns, Brooks & Spivak, in press). In addition, we computed the scan-to-scan movement from the AFNI motion files in all three principal directions during exposure to each source (handler, stranger, computer) and condition (reward and no-reward) for each dog for all volumes, including those subsequently censored. Scan-to-scan movement was computed as: sqrt(dxi2+dyi2+dzi2) where dxi, dyi, and dzi are the changes in corresponding head position of the ith volume (e.g., dxi = xi−xi−1). Scan-to-scan movements greater than 10 mm occurred occasionally if a dog moved out of the field of view. In these cases, the motion estimates were not reliable and so these values were capped at 10 mm to avoid biasing the average. We also computed the proportion of censored volumes during exposure to each source and condition. This allowed us to determine whether unbalanced motion parameters or censoring between reward and no-reward conditions might introduce strong bias into our BOLD findings.

EPI images were smoothed and normalized to %-signal change. Smoothing was applied using 3dmerge, with a 6 mm kernel at Full-Width Half-Maximum (FWHM). The resulting images were then input into the General Linear Model.

For each subject, a General Linear Model was estimated for each voxel using 3dDeconvolve. The task-related regressors in this model were: (1) familiar human reward signal, (2) familiar human no-reward signal, (3) unfamiliar human reward signal, (4) unfamiliar human no-reward signal, (5) computer reward signal, and (6) computer no-reward signal. Because our previous work measuring the hemodynamic response function (hrf) in dogs on this task revealed a peak response at 4–6 s after signal onset (Berns, Brooks & Spivak, 2012), the six task regressors were modeled as impulse functions. Events were convolved with a single gamma function approximating the hrf. Motion regressors generated by the motion correction were also included in the model to further control for motion effects. A constant and linear drift term was included for each run. Finally, to control for possibly confounding physiological factors (e.g., if the reward condition led to more rapid breathing, which might boost neural BOLD signal), a spherical ROI (3 mm radius) was drawn manually on each dog’s structural image in the posterior ventricle, just posterior to the splenium of the corpus callosum. Using AFNI’s 3dmaskave, average timecourses for these ROIs were then extracted after transforming to each dog’s structural space. This timecourse was entered into the general model as a nuisance variable for each subject. Because BOLD signal changes in CSF and white matter reflect physiological effects, not neural processing, this controlled for undue influence of physiological effects on the primary contrasts (Weissenbacher et al., 2009; Murphy, Birn & Bandettini, 2013).

ROIs and mixed-effects modeling

To measure the interaction of reward and no-reward signals with signal source, we used a mixed-effects ANOVA to compare mean caudate activation in the six conditions (2 signals × 3 sources) across all dogs. The four primary temperament factors (as computed from the CBARQ behavioral questionnaire and subsequent PCA discussed above) were also included in the statistical model. To ensure that we were, in fact, measuring caudate activity, we used anatomically defined ROIs. A left and right ROI was drawn on each dog’s structural image (Fig. 3). Then, these ROIs were used to extract average beta values for each dog in each condition from the first-level GLMs, after transforming to each dog’s structural space. The end result was 12 values for each dog (the 6 conditions for both left and right caudates).

Figure 3 The right caudate seed, anatomically defined, used for participant Kady, in the transverse (A) and coronal (B) planes.

Analysis was conducted using the Mixed Models procedure in SPSS v21 (IBM). A 2 × 3 ANOVA was formulated with fixed effects for hand signal (reward, no reward) and source (familiar, unfamiliar, computer) with dog as a random effect. Because no significant left/right differences were observed, side was not included as a fixed effect. We also conducted a second analysis that included the C-BARQ temperament factors as well their interactions with the other fixed effects. This allowed us to determine the relative caudate activation in each of the six conditions and examine how this might be modulated by temperament.

PPI

In addition, we used our data to compute a psychophysiological interaction (PPI) analysis at the individual and group levels. PPI highlights areas that increase functional coupling (defined as synchronous firing patterns) with a seed area in a certain condition, or in a certain condition in contrast to some other (Friston et al., 1997), and has been shown to produce reliable and robust measures of task-specific functional connectivity (Kim & Hortwitz, 2008). PPI allows one to determine which brain areas increase functional connectivity with an area of interest during processing of a specific task.

We computed our PPI in AFNI as described in http://afni.nimh.nih.gov/sscc/gangc/CD-CorrAna.html. First, functionally localized, bilateral caudate seeds were generated for each dog from the allrew–allnorew contrast (that is, across all three source conditions: familiar, unfamiliar, and computer). For each animal, the seeds were generated as spheres with a 3 mm radius centered on the voxel with the greatest differential activation in the left and right caudate respectively.

The activation time series for the left and right caudate seeds were extracted for each subject using 3dmaskave. The time series were then deconvolved using 3dDetrend using a basic gamma function. Next, the interaction of the condition times (reward, no reward, and neutral baseline) and the deconvolved caudate seed time series was computed to produce an interaction term. The interaction term was convolved using the AFNI waver command with a basic gamma function. The convolved interaction term and the initial extracted caudate seed time series were then entered as regressors into the initial GLM (along with the primary task regressors, drift terms, motion regressors), and physiological regressor).

Increased BOLD activation corresponding to the interaction term in the GLM should then be seen in areas that increased functional coupling (i.e., synchronous firing) with the caudate during reward versus no-reward conditions. Such areas may be understood to be differentially connected to the caudate during processing of the reward signal in this task. Note that, as a connectivity analysis, PPI is driven by patterns of increase and decrease in BOLD signal, and is not necessarily indicative of a main effect difference in BOLD strength in any particular region.

Whole-brain group analyses

To apply the transformations to a statistical contrast, the appropriate individual-level contrast (obtained from the GLM as described above) was extracted from the AFNI BRIK file and normalized to template space. Group normalization was conducted using the Advanced Normalization Tools (ANTs) software, as described in Berns, Brooks & Spivak (in press). Briefly, three spatial transformations were computed for each dog: (1) rigid-body mean EPI to high-resolution structural (6 dof); (2) affine structural to template (12 dof); and (3) diffeomorphic structural to template. Spatial transformations were then concatenated and applied to individual contrasts obtained from the above-described GLM model. This allowed the computation of group level statistics. For group-level statistics, a high-resolution MRI beagle brain atlas was used as the template (Datta et al., 2012). AFNI’s 3dttest+ + was then used to compute a t-test across dogs with the null hypothesis that each voxel had a mean value of zero. All twelve dogs were used in this group analysis. The group contrast we conducted was reward—no reward (across all source conditions), computed as the contrast [(Rewfam + Rewunf + Rewcom)]−[(noRewfam + noRewunf + noRewcom)]. The same approach was taken to compute the group PPI results. Instead of a contrast, the beta values for the interaction term (computed between stimulus presentation schedule and caudate time series) from each individual dog were entered into a second-level model using 3dttest+ +. For both the primary GLM and the PPI analysis, we then calculated the average smoothness of the residuals using 3dFWHMx and then used 3dClustsim to estimate the significance of different cluster sizes across the whole brain after correcting for familywise error (FWE).

Results

Mean scan-to-scan movement across all sources (familiar and unfamiliar human and computer) and signals (reward and no-reward) was 0.94 mm. In a repeated measures ANOVA, source was a significant predictor of motion (F(2) = 44.6, p < 0.001), with motion lower in the familiar handler source than the other two. Signal, however, was not a significant predictor of motion (F(1) = 0.9, p = 0.366), nor was the interaction of source and signal (F(2) = 0.08, p = 0.928). This paralleled the findings from a repeated measures ANOVA examining the percentage of volumes censored across all source and signal conditions. In the latter, source was a significant predictor of censoring (F(2) = 16.2, p < 0.001), while signal was not (F(1) = 1.0, p = 0.333), nor was the interaction of source and signal (F(2) = 2.0, p = 0.156).

Whole-brain group analysis of (reward–no-reward) hand signals collapsed across all three source conditions (familiar human, unfamiliar human, and computer) yielded robust and significant bilateral ventral caudate activation. With a single-voxel significance of 0.005, the cluster, which corrected FWE across the whole brain, was p = 0.01 (Fig. 4).

Figure 4 Whole-brain group analysis of response to all reward–no-reward conditions.

An unthresholded transverse slice (A) and coronal slice (B) are shown, as is a coronal slice thresholded at 0.005 (C). Color indicates t-statistic at each voxel against the null hypothesis of equal activity to reward and no-reward conditions. Significantly greater activity was observed in the reward versus no-reward condition.

In the four-factor PCA of the 14 temperament factors from the C-BARQ questionnaire (Table 2), the first factor accounted for 31.1% of variance, the second for 17.5%, the third for 15.0% and the fourth for 13.3% of variance (accounting cumulatively for 76.9% of total variance). Of note, the first factor appeared to strongly represent aggressivity, while the second was associated with attachment and separation.

Table 2 Weighting of C-BARQ PCA factors.

Temperament	Component 1	Component 2	Component 3	Component 4	
Stranger-directed aggression	.705	.373	−.319	−.161	
Owner-directed aggression	.882	−.174	−.383	.114	
Dog-directed aggression	.735	−.104	.416	−.209	
Dog-directed fear	.450	−.153	.686	.451	
Familiar-dog aggression	.887	−.323	.082	.123	
Trainability	−.111	.043	.805	−.245	
Chasing	.785	.217	.260	.130	
Stranger-directed fear	−.016	.028	.290	.801	
Nonsocial fear	.577	−.396	.106	−.317	
Separation-related problems	.237	.716	.044	−.348	
Touch sensitivity	.580	.149	−.571	.405	
Excitability	.341	.613	.013	−.115	
Attachment/attention seeking	.097	.825	.250	−.150	
Energy	−.270	.587	.016	.654	

In the mixed-effects ANOVA model without temperament included, the interaction of source and signal was not a significant predictor of caudate activation (F(2, 67.7) = 1.619, p = 0.206). However, in the mixed-effects ANOVA model with the four temperament factors included, signal was a significant predictor (F(1, 88.6) = 19.3, p < 0.001), the interaction of source and signal was a significant predictor (F(2, 61.8) = 3.6, p < 0.05), the interaction of source and C-BARQ factor 1 was a significant predictor (F(2, 61.8) = 11.6, p < 0.001), the interaction of source with signal and C-Barq factor 1 was a significant predictor (F(2, 61.8) = 29.9, p < 0.001), the interaction of source with C-BARQ factor 2 was a significant predictor (F(2, 61.8) = 13.6, p < 0.001), the interaction of source with C-BARQ factor 3 was a significant predictor (F(2, 61.8) = 4.9, p = 0.01), and the interaction between source, signal, and C-BARQ factor 3 was a significant predictor (F(2, 61.8) = 19.3, p < 0.001). Neither C-BARQ factor 4, nor any of its interactions, were significant predictors of caudate activation. For further analysis we focused on C-BARQ factor 1. C-BARQ factor 2 did not show a significant interaction with signal, and C-BARQ factor 3 was dominated by the trainability metric from the C-BARQ questionnaire. The values for our dogs showed very limited range in this metric, and did not match with independent assessments of actual trainability by the authors (two of whom have extensive animal training experience) (Fig. 5).

Figure 5 Activation within caudate ROIs in dogs with low and high aggressivity (relative to our sample mean) for reward and no-reward signals across the three source conditions.

Values and s.e. are derived from the full mixed-effects model, using Z-scores for C-BARQ factor 1 of + 1 for dogs with higher aggressivity and −1 for dogs with lower aggressivity (±1 corresponded to the upper and lower limits of the scores). Dogs with lower aggressivity showed significantly greater caudate activation to reward versus no reward signals presented by their familiar handler (p < 0.001), but not from unfamiliar humans (p = 0.15) or computers (p = 0.09), while dogs with higher aggressivity showed significantly greater activation to the reward versus no reward signals from the unfamiliar human (p = 0.003) and computer (p < 0.001), but not their familiar handler (p = 0.07).

Functional connectivity group analysis revealed bilateral clusters of increased activation, in the left posterior suprasylvian region corresponding to the right caudate seed, and in the right posterior suprasylvian region corresponding to the left caudate seed (Fig. 6). Both clusters were significant at voxelwise p < 0.05 (321 voxels for the left posterior suprasylvian region, 565 voxels for the right posterior suprasylvian region), but do not survive thresholding with whole-brain corrected FWE at p < 0.10.

Figure 6 Whole-brain group analysis of the interaction between BOLD time course in the left and the right caudate seeds and signal presentations—warmer colors here represent increased functional coupling with the caudate seed during presentations of the reward versus no-reward signals across all three source conditions (familiar human, unfamiliar human, computer).

The cluster corresponding to the right caudate seed ((A), upper left unthresholded and (C), lower left thresholded voxelwise at p < 0.05) is in the left posterior suprasylvian region, and the cluster corresponding to the left caudate seed ((B), upper right unthresholded and (D), lower right thresholded voxelwise at p < 0.05) is in the right posterior suprasylvian region. Color indicates t-statistic at each voxel against the null hypothesis of equal connectivity to the caudate for reward and no-reward conditions.

Discussion

Here we showed that, across 12 dogs, the caudate was differentially active for reward vs. no-reward signals when analyzed at the group level. Findings controlled for confounding effects from motion, censoring, and physiological changes. This builds on our previous findings showing differential caudate activation for a similar task at the individual subject level (Berns, Brooks & Spivak, 2012; Berns, Brooks & Spivak, 2013). Given this robust group finding, and the extensive literature linking caudate activation to reward anticipation (Montague & Berns, 2002; Schultz, Dayan & Montague, 1997; Knutson et al., 2001), the current experiment provides the most definitive evidence to date that fMRI with unrestrained, awake dogs can yield reliable and valid data. Moreover, the current results emphasize the importance of the source of information to the dogs and how this interacts with their temperaments.

Our primary interest in the current experiment was to explore the effect of signal source on reward processing and the extent it matters to a dog whether a reward or no-reward signal comes from a familiar or an unfamiliar human, or from a human or a computer. If behavioral and neural response to reward signals are merely products of specific reinforcement history, dogs should not strongly differentiate signals with the same meaning across different source conditions. Differential caudate activation is a reliable way to probe these questions, and can be interpreted as a marker of positive salience even in the absence of a specific behavior (Ariely & Berns, 2010; Bartra, McGuire & Kable, 2013). Brain data now confirms prior behavioral evidence indicating that dogs can generalize meaningful signals when produced by unfamiliar humans, and that they can learn and respond to meaningful signals produced by computers.

When examined in a mixed-effects model including signal type and signal source, the difference between reward and no-reward signals in the caudate was not significantly different across the three sources. However, including a factor representative of a key attribute of canine temperament (aggressivity) revealed significant interactions between signal source and temperament. In other words, signal source does matter to dogs, and apparently quite strongly—but the way in which it matters is highly dependent on the dog’s temperament. Specifically, dogs with lower aggressivity showed a higher differential caudate response to reward versus no-reward signals from their handlers, while dogs with higher aggressivity showed a higher differential caudate response to reward versus no-reward signals from the unfamiliar humans and the computer. It must be noted that “low” and “high” aggressivity measures here are relative to our sample—none of our subjects scored particularly high on C-BARQ aggressivity measures. Moreover, the differences in caudate activation were not due simply to changes in physiological arousal as these were controlled by the inclusion of a physiological proxy vis-à-vis an ROI in the CSF. Nor were the caudate differences due to motion because there was no significant difference in scan-to-scan motion of reward vs. no reward signals across the 3 sources. Interestingly, the differences in caudate activation were not correlated with the C-BARQ factors for attachment and separation.

The interrelation of individual differences and neurological and behavioral responses is foundational to contemporary human psychology (e.g., Depue & Collins, 1999; Ajzen, 2005). Although less studied, there is still substantial work examining temperament in non-human animals (Gosling & John, 1999), much of it recently in dogs (Jones & Gosling, 2005; Taylor & Mills, 2006; De Meester et al., 2011; Dowling-Guyer, Marder & D’arpino, 2011), and strong evidence indicates that, just as in humans, temperament is an important factor affecting neural and behavioral response in different contexts. Our current finding, that the caudates of dogs with lower aggressivity respond more strongly to reward versus no-reward signals from familiar handlers while those of dogs with higher aggressivity respond more strongly to unfamiliar humans, is in line with prior literature on striatal reward processing. Striatal response to reward depends heavily on salience (Zink et al., 2004), and anxiety predisposes one to attend to possible threat (MacLeod & Mathews, 1988). Dogs who show higher aggressivity may be more aroused or more anxious in the presence of a stranger than with a familiar handler, and this likely increases the salience of the unfamiliar human, and thus the striatal activation to reward. Dogs with lower aggressivity, on the other hand, may find their owner relatively more salient due to prior history of interaction and reward.

As a side note, we cannot fully rule out differences in training approaches used by individual handlers as a possible contributor to our findings—it is plausible that dogs with different temperament may elicit different training approaches from their handlers. However, all dogs were trained to a similar criterion of task success using the same general approach, with biweekly oversight by project staff, over roughly the same amount of time. Because the training for this task was, at essence, very simple (repeated exposures leading to classical conditioning of the reward and no reward cues), it seems unlikely that slight differences in training would account for the results reported here.

Due to our relatively large dataset (90 10 s trials per dog across all three source conditions) we were also able to conduct a PPI connectivity analysis to look for brain regions that increased functional coupling with the caudate at the group level on reward versus no-reward trials. Functional connectivity analysis with the right and left caudate seeds highlighted contralateral cortical patches (right for the left caudate seed and left for the right caudate seed) in the posterior suprasylvian region. There is no prior work on this region in canines, but there is a substantial literature in cats, who, as carnivores, are fairly closely related to dogs. Evidence suggests the posterior suprasylvian region in carnivores is a downstream visual area necessary for learning and discriminating between novel visual stimuli (Markuszka, 1978; Updyke, 1986; Lomber, Payne & Cornwell, 1996). The posterior suprasylvian region in cats is functionally analogous to inferotemporal cortex in primates, which is also shown to play an integral role in learning new visual associations, including for faces (Horel et al., 1987). In addition, the posterior suprasylvian region in cats and the inferotemporal region in primates have been shown to share strong connections with the striatum (Yeterian & Hoesen, 1978; Royce, 1982; Webster, Bachevalier & Ungerleider, 1993). Given this, a possible interpretation of our current connectivity findings is that the caudate and posterior suprasylvian regions differentially coupled contralaterally to support contextual visual learning related to reward anticipation. This is particularly likely given that testing occurred in a novel environment, and, in the case of the stranger source condition, with a novel signal giver (likely with their own slight idiosyncrasies in signal presentation). In other words, visual features of the signal and signal presentation at MRI were subtly different from those previously experienced by our subjects, likely leading to additional learning about what contexts, signals, and signal givers might lead to reward. Due to the fairly low statistical threshold of these findings, they should be taken as descriptive and suggestive as opposed to conclusive, but, due to the lack of connectivity data in canines, are still of interest.

In brief, we demonstrated a robust bilateral differential caudate activation to reward versus no reward signals at the group level in 12 dogs across three source conditions. Further, a measure of temperament, specifically aggressivity, was a strong predictor of differential caudate response across the three source conditions. A condition-specific functional connectivity analysis indicated increased contralateral coupling between the right and left caudate and visual learning brain regions. These findings provide new understanding of reward processing in the domestic dog and contribute to a growing body of research on individual differences in non-human animals. Particularly notable is the explanatory power of a temperament measure in explaining the neural response. It is likely that there are substantial individual differences in how different dogs will react across a range of contexts, and future research and applied work should be sensitive to this, particularly when making broad claims based on findings across a group of dogs whose temperament has not been assessed. BOLD signal in the caudate may serve as a predictive measure of dog temperament and amenability to different training approaches, although, due to the difficulty and expense of fMRI, application in the near term would likely be restricted to special instances (e.g., assessment of military or service dogs).

We are grateful to all of the dogs’ owners for the time they have devoted to training: Lorraine Backer (Caylin), Cindy Keen (Jack), Patricia King (Kady), Claire Mancebo (Libby), Jeff Petermann (Nelson), Cecilia Kurland (Ohana), Vicki D’Amico (Pearl), Nicole Zitron (Stella), Aliza Levenson (Tigger), Lisa Tallant (Velcro), Darlene Coyne (Zen), and GB’s dog, Callie. Thanks to Helen Berns for developing the treat-kabob and dog photos.

Additional Information and Declarations

Competing Interests

Author Contributions

Animal Ethics

Gregory Berns and Mark Spivak own equity in Dog Star Technologies and developed technology used in the research described in this paper. The terms of this arrangement have been reviewed and approved by Emory University in accordance with its conflict of interest policies. Mark Spivak is an employee of Comprehensive Pet Therapy. Peter Cook has no competing interests.

Peter F. Cook conceived and designed the experiments, performed the experiments, analyzed the data, contributed reagents/materials/analysis tools, wrote the paper, prepared figures and/or tables.

Mark Spivak conceived and designed the experiments, performed the experiments, wrote the paper, reviewed drafts of the paper.

Gregory S. Berns conceived and designed the experiments, performed the experiments, analyzed the data, contributed reagents/materials/analysis tools, wrote the paper, prepared figures and/or tables, reviewed drafts of the paper.

The following information was supplied relating to ethical approvals (i.e., approving body and any reference numbers):

This study was performed in strict accordance with the recommendations in the Guide for the Care and Use of Laboratory Animals of the National Institutes of Health. The study was approved by the Emory University IACUC (Protocol #DAR-2001274-120814BA), and all dogs’ owners gave written consent for participation in the study.

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
