# Peer review of "One pair of hands is not like another: caudate BOLD response in dogs depends on signal source and canine temperament"

_PeerJ, doi:10.7717/peerj.596_

## Round 0.1 · original submission · Major Revisions

Please consider the criticisms on the methodological sides by Reviewer 2 and 3, and the opportunity to be more cautious in your conclusions as suggested by Reviewer 2.

·

Basic reporting

The article is well written, with sufficient introduction and references.
The figures are relevant to the content and well described and labeled, although some of them have a bad resolution (figure 3, figure 4 and figure 6).

Experimental design

Material and methods are very good described, leaving no place for any doubtful point. Each point of the experimental design is thoroughly described and documented with valid references. I appreciate the attention put on dogs’ welfare. Nevertheless, I found this section too long and the sample rather poor.

Validity of the findings

Discussion and conclusions are very interesting for behavioural medicine. It is very interesting and useful the connection between dogs’ temperament and neural response. Nevertheless, I don’t believe in fMRI as a means for dogs temperament evaluation in every day practice, due to the hard training that dogs must follow

Additional comments

You can also cite “Siniscalchi et al., Seeing Left- or Right-Asymmetric Tail Wagging Produces Different Emotional Responses in Dogs, Current Biology (2013), (line 65) and “Siniscalchi M, Sasso R, Pepe AM, Vallortigara G, Quaranta A, Dogs turn left to emotional stimuli, Behavioural Brain Research 208 (2010) 516–521” (line 66).
There is a spelling mistake in line 65 and in line 609: the exact name is “Quaranta” (not Qauranta).

Reviewer 2 ·

Basic reporting

Building on their previous fMRI work, Cook et al., present data suggesting a correlation between caudal nuclei fMRI activation and reward signals presented in the form of a familiar or unfamiliar stimulus, in awake unrestrained dogs. The manuscript is well written, and the results of this work are novel and of potential interest for a broad readership.

Experimental design

See below

Validity of the findings

I have several major concerns about the fMRI procedure employed by the authors in the present and previous studies which I detailed below:

- The authors report the ability to train the dogs to keep motionless for 30 seconds, but then acquire fMRI timeseries over a 6 min time window. As a result, the procedure heavily relies on operator-based “censoring” of the inevitable motion artefacts occurring during the fMRI scans, resulting in a rejection rate of as much as 50% (!) of the fMRI volumes. This is not customary in animal or human fMRI imaging as it obviously could introduce enormous bias in the generation and interpretation of results. This procedure per se casts doubts on the reliability of the signal mapped, independent on the authors’ claims in this and their previous work.

- To my surprise, no attempt to control for peripheral variables that can greatly affect BOLD fMRI signals has been described by the authors, in this or any of their previous work. These include for example parameters such as pCO2 which can be monitored non invasively and whose dynamics can greatly influence central BOLD responses. Importantly, arousal states, such as those associated to the tasks can induced hyperventilation, resulting in rapid changes in CO2 levels that can affect BOLD signal. Similarly, arousal-related peripheral changes in blood pressure could also affect central signal changes. The authors completely neglected such important factors which should at least be discussed.

- Another major source of potential bias is the lack of high resolution anatomical images co-centerd with the EPI scans. Again, the approach used by the authors (disjunct anatomical vs EPI) is not customary in animal or human fMRI research and it should be commented.

- The authors have opted for the unusual choice of showing unthresholded group-statistics maps. They should elaborate on the reason for such as an unusual choice and should included thresholded maps for the readers' perusal. It is also not clear whether the authors included in their maps family-wise corrections for multiple comparisons, and if not they should include as supplementary data a thresholded and cluster corrected map.

- A more detailed explanation of the PPI approach should be included in the work for non fMRI specialists

Additional comments

While the results of the study are interesting, I believe the experimental approach employed is prone to major confounds and bias, which would probably make the work unpublishable in specialist neuroimaging journals. I believe the authors should greatly tone down their claims, and substantially rewrite the manuscript to openly acknowledge these major methodological flaws.

Reviewer 3 ·

Basic reporting

General comments

The current manuscript assesses the importance of signal source type to dogs’ caudate activation and its association with canine temperament. It is an interesting extension of previous research in the area of dog-human social interaction.

-This is an interesting and well-written article that is worthy for publication. However I have concerns mostly with the omission of some C-BARQ data on dogs’ temperament, which could partially extend the results and improve the paper in the whole, see my specific comments below.

-Uses of the general terms "aggressive" and "calm" in the current manuscript are quite loose, and sometimes seem to be based more on colloquial definitions rather than scientific definitions. Given that the dogs’ temperament measuring instrument (i.e. the C-BARQ) that was used assessed the levels of stranger-directed aggression, it might be more appropriate to change the terminology with "high" and "low" stranger-directed aggression in order to well reflect the two behavioral categories.

There are many kinds of aggressive behavior among dogs and it is not said that dogs with higher levels of stranger-directed aggression also display other forms of aggression; on the other hand, dogs with low levels of stranger-directed aggression (“calm” dogs in the paper) could be very nervous!

-The Introduction is a well-written section, however perhaps a little bit too condensed.
Page 5, first paragraph:” Although the subject dogs had an extensive reinforcement history with the human and computer signals, we hypothesized… in line with the possibility that social bond, and not just food-specific reinforcement history, affects the valence of familiar cues.”:

There is now evidence for the presence of a correlation between the owners' attachment profile and the owner-dog attachment bond" which could support this statement, the following reference should be added at this point and discussed also in the light of the main topic of the paper (i.e. the sensitivity of the domestic dog to human social interaction): Siniscalchi M, Stipo C, Quaranta A (2013) "Like Owner, Like Dog": Correlation between the Owner's Attachment Profile and the Owner-Dog Bond. PloS ONE 8(10).

-Why dogs did not received any type of reinforcement history also with the unfamiliar-human signals?

-Figure 5 is appropriately described even if the use of different numbers of asterisks could be useful in order to make the level of significace clearer to the reader (e.g. * P<0.05; ** P<0.01; ***P<0.001); behavioral categories labels “calm” and “aggressive” should be changed, – see previous comments.

Experimental design

-Methods – experimental design

-Page 12:”The six runs for each dog were always in the same order: familiar human, unfamiliar human, computer, computer, unfamiliar human, familiar human. Low sample size didn’t allow for reasonable counterbalancing across participants, and this ABCCBA pattern of presentation controlled for effects in the BOLD signal from habituation, sensitization,…”.

I agree that counterbalancing was impossible due to the low sample size but why the six runs were not presented randomly /semi randomly at daily-weekly intervals? ABCCBA would be good if dogs did not get habituated to the testing paradigm (reward no reward) or if the habituation is linear and constant over trials (e.g. ABCCBA = 654321 → A total:6+1=7; B total:5+2=7; C total:4+3=7; different numbers parallel different levels of response to stimuli) but if subjects get habituated quickly, results could be affected by habituation - e.g. ABCCBA = 642111 -> A total=7; B total=5; C total=3); in my experience, when dogs are tested in labs, they usually get quickly habituated (and tired) to the testing paradigm (indipendently of the type of stimulus), and as a consequence, we prefer testing them with repeated randomly-semi randomly runs at daily or weekly intervals. In addition, when food is used as reward the satiety levels could also influence dogs’motivation to achieve tasks effectively over trials. All of these aspects could partially explain the differences in the activation within caudate ROIs across the three source conditions (due to habituation), at least for the “calm” dogs categoriy (see figure 5). I suggested a few control analyses to be executed for clearing up this situation, if this seems to be too difficult, I suggest at least to incorporating this hypothesis to the discussion.

-Page 12, line 7:” In addition, we determined that placing the computer runs first or last would be difficult for some dogs.” How did the authors determine that? Is it just a hypothesis? Pilot tests?

“…by placing them in the middle,… but also had the more familiar (potentially easier) source conditions at the end of the experimental sequence, when stress and fatigue may have made them less inclined to continue participating”

These differences (i.e. possible differences in the % of response to different stimuli) should be revealed from results and not pre-determined in the experimental design…

Page 13

-C-BARQ
Second paragraph:” Because of the introduction of the unfamiliar human in our testing protocol, we were primarily interested in the stranger-directed aggression subscale.”

Since the dog’s temperament is one of the main topic of the paper (as it comes up even from the title) I think that the analysys of others C-BARQ behavioral categories (e.g. owner-directed aggression, separation-related behaviour, attachment or attention-seeking behaviour, excitability, etc.) must be crucial for extending the results and for improving the general understandig of the paper.
It should not be extremely difficult to extend the behavioral analysis since all dogs’ handlers completed the C-BARQ and consequently most of the data are already available.

-Third line from the bottom:” Raw scores were converted to Zscores based on our sample mean (0.34) and standard deviation (0.45) to produce behavioral metrics usable in mixed-effects modeling."

The statistical procedure here should be explained in a more detailed fashion.

Discussion
-Page 22 - first paragraph:” This could explain our current finding: dogs who show higher stranger aggression are most likely more anxious in the presence of a stranger than with a familiar handler, and this likely increases the salience of the unfamiliar human. Calm dogs, on the other hand, may find their owner more salient due to prior history of interaction and reward.”.

This is not very clear: if dogs are more anxious in the presence of a stranger they could be also more distracted by his presence; on the other hand, if it is true that anxious dogs perform better in this task, what about the possibility that calm dogs have higher levels of owner-directed aggression? More behavioral analysis is required – see previous comments.

Page 22, Line 16 from the top:” In the cat, the primary head and face cortex is anterior and superior, while the rest of the sensory homunculus ..”

Please change “homunculus” with “felunculus” which is the appropriate term.

Validity of the findings

This is strictly related to the experimental design - see previous comments.

Additional comments

The use of functional MRI is an emerging and fascinating technique to explore dog cognition but the experimenters should be more aware about the difficulties in avoiding possible variables during testing rewarding systems using this method. These are originating from the possible (and likely) effect of training program to prepare dogs to be comfortable with the physical confines of the MRI that required dogs to be rewarded step by step (behavior shaping, desensitization, habituation and behavior chaining). “Dogs were cleared for testing when they were able to do an extended “dress rehearsal” session in the mock scanner without showing anxiety or escape behavior “– but this was possible because they were waiting a reward in the meanwhile…

---

## Round 0.2 · accepted · Accept

I concur with the Reviewer's comment and I believe the paper was reviewed adequately.

Reviewer 2 ·

Basic reporting

No comments

Experimental design

The authors adequately addressed my methodological concerns.

Validity of the findings

The authors adequately addressed my methodological concerns.